# Processing Non-Gaussian Data Residuals in Geomagnetism

**Andrey Khokhlov** [1,2]

1 Geophysical Center RAS, 3, Molodezhnaya St., 119296 Moscow, Russia; fbmotion@gmail.com
2 IEPT Russian Academy of Sciences, 84/32, Profsoyuznaya, 117997 Moscow, Russia

**Featured Application: A possible application of the data processing method described below is to separate noise due to external effects from noise due to the limited accuracy of the sensor itself. Such a model may reveal the need for a calibration process and impose some statistical constraints on external effects.**

**Abstract:** Some time ago, we considered non-Gaussian shapes of histograms of quantities that were related to residuals in data: we showed at a qualitative level that non-Gaussianity is most likely the result of mixing of Gaussian distributions. In this addendum, we argue that there is a quantitative description that can be used in fairly general situations. Briefly, we present here the same magnetic measurement data that were reported in the original publication: Khokhlov, A.; Hulot, G. On the cause of the non-Gaussian distribution of residuals in geomagnetism. *Geophys. J. Int.* **2017**, *209*, 1036–1047.

**Keywords:** probability distributions; magnetic field variations through time; data modeling; satellite magnetic measurements

## 1. Introduction

Observations of long-term natural processes, in particular magnetometry, at first sight, fully meet the initial requirements of mathematical statistics and, thus, the methods of this scientific discipline are widely used in data processing. At the same time, the characteristic of deviation (from mean values, baseline, mean scatter of instrument readings, etc.), which is usually referred to as the residual term, is important enough for estimating the quality of observations. Corresponding statistics are related to the simplest statistical formula, which works fine in theory in the situation of fairness of the Gaussian hypothesis for the recorded data. The same hypothesis, at first sight, should be fulfilled in the situation of vector magnetic measurements because the observed value of the magnetic vector is always a sum of magnetic influences from many local and global sources, and the distribution law for sums of random variables is predicted by the Central Limit Theorem of Probability Theory, which states that under rather simple conditions, the sum of many independent random influences is very well approximated by the Gaussian law.

Therefore, it is a common opinion that in processing observational data and calculating the characteristics of these data (for example, calculating the mean value of the residual), we can use Gaussian statistics without any restrictions and there is no reason to doubt the numerical value of the answer, which corresponds to the RMS deviation $\sigma$ of a Gaussian random variable. However, the stationary nature of natural fluctuations can be assumed only with respect to specific time intervals: magnetic field measurements over a relatively short time can include significant nonstationarity of external factors—seasonal variations, magnetic storms, the presence of specific technological disturbances, etc. Thus, the idea that the observational series is just a sample of the general population corresponding to a fixed Gaussian random variable will generally give a very unrealistic picture. A more adequate model seems to be that of a mixture of random variables differing in parameters, and

parameter variations are described by a separate random variable whose value distribution can be neglected only in a stationary situation.

Earlier, the general approach was outlined in the article [1] for the example of real magnetic data, referring, however, not to ground-based magnetic observations, but to observations of magnetic satellites of the project Swarm, crossing different intensities of magnetic fields. In this case, in a relatively small time interval, it is possible to collect a lot of data that reflect random field variations, the corresponding data model was substantiated in detail in [1].

In the present article, we will consider the computations and, for the convenience of understanding the statistical model of variability, we will refer to exactly the same data as in [1]. The difference in the method of processing a large volume of SWARM magnetometer data and a large volume of observatory magnetometer data is insignificant for the explanation of the method as such.

Recall that typical appropriate statistical assumptions are being made with respect to the distribution of the uncertainties $\sigma_i$ affecting the data used. The statistical properties of these uncertainties, however, are not always well characterized. In such circumstances, assuming that uncertainties follow a Gaussian distribution would a priori make sense, since such a distribution often arises naturally as a consequence of the central limit theorem (i.e., when errors act in an additive manner). Relying on this assumption, and provided that $s_i$ is an adequate measure of the error affecting the datum $\gamma_i$, standard statistical estimations are then used to infer the model. The normalized residuals $\left\{ \frac{\gamma_i - \widehat{\gamma}_i}{s_i} \right\}$ (here $\widehat{\gamma}_i$ being the datum value predicted by the model) are expected to follow a standard normal distribution.

Yet, residuals of geomagnetic observations of an arbitrary type often display a sharper distribution, sometimes much closer to that of the so-called Laplace distribution (e.g., [2–5]). We suggested in [1] that residuals may be incorrectly normalized and, therefore, their common statistical distribution is a mixture of Gaussian distributions. In particular, we demonstrated in [1] several examples of the variability in $\sigma$ determination that indeed lead to the non-Gaussian shape of the histogram. Thus, we assume that the observable residual $\theta$ is a mixture of individual Gaussian random variables with zero expectations and random variances $\beta^2$. In the present short note, we argue that the distribution of the random variable $\beta$ can be well approximated by the lognormal distribution with pdf

$$ f_\beta(t) = \frac{1}{ts\sqrt{2\pi}} \exp\left[ -\frac{\ln^2 t}{2s^2} \right] $$

We also provide the method that recovers the value of $s$ in the real data case.

## 2. Mixture Model

### 2.1. The Unformal Interpretation

The mixture model is appropriate for the situation when the data are inhomogeneous, for instance, they come from several locations such that each region slightly perturbs the assumed data distribution law, i.e., the corresponding distribution formulae differ slightly in their parameters. In practice, we often face an even simpler situation: each set of regional data is Gaussian with mean zero but the corresponding $\sigma$-value depends on the region. However, we rarely can select the region with absolutely homogeneous data in it; therefore, we better simulate these situations by means of sequential small perturbations of the initially homogeneous Gaussian population. Can the limit distribution be described given the very small intermediate perturbations?

Of course, the model of successive repeated small random perturbations applied to a stationary process is only one of many possible ones. However, this model allows statistical estimation of residuals in a simple and straightforward computational procedure and eventually allows the model data to be simulated in order to compare them with the actually observed data. Relevant comparison procedures are available in nonparametric statistics, such as the well-known Kolmogorov–Smirnov test.

The statistical Kolmogorov–Smirnov criterion on large samples is very sensitive and therefore can certainly indicate more subtle effects, even of non-random nature, present in the data.

### 2.2. Version of the General Formula

If $\zeta$ is an arbitrary random variable with density $f_\zeta$, then for fixed $y_0 > 0$, the ratio $\zeta / y_0$ has density $f_\zeta(xy_0)y_0$, see also (see, e.g., [6]). Let now the denominator not be fixed but a positive random variable with density $g_\eta$, then, we obtain the pdf for this ratio

$$h(x) = \int\limits_0^{+\infty} f_\zeta(xy) y g_\eta(y) dy$$

We may now compare the mixture of unbiased Gaussian distributions (i.e., with pdf $f_\alpha = \mathcal{N}(0, \sigma^2)$) by randomizing their standard deviations using a random variable $\beta > 0$:

$$f_\theta(x) = \frac{1}{\sqrt{2\pi}} \int\limits_0^\infty \exp\left(-\frac{1}{2}\frac{x^2}{t^2}\right) t^{-1} f_\beta(t) dt = \int\limits_0^\infty \frac{1}{\sqrt{2\pi}} \exp\left(-\frac{1}{2}y^2 x^2\right) y g_\eta(y) dy$$

Obviously, $f_\theta$ can be interpreted as the pdf of the ratio.

For instance, recall the following example of [1]: the uniform mixture of unbiased Gaussian distributions with standard deviations varying between 0 and 1, i.e., mixing pdf is

$$f_\beta(t) = \begin{cases} 1 & 0 < t < 1 \\ 0 & \text{otherwise} \end{cases}$$

We may treat that mixture as the ratio of standard Gaussian $\alpha$ divided by $\eta$—the inverse of the uniform distribution:

$$g_\eta(y) = \begin{cases} \frac{1}{y^2} & y > 1 \\ 0 & y \leqslant 1 \end{cases}$$

### 2.3. Sequential Small Mixtures

The *small* multiplicative randomization is described in terms of a random $\beta > 0$ with pdf $f_\beta \sim 0$ out of $[1 - \varepsilon, 1 + \varepsilon]$ for some small $\varepsilon$, we may assume $\beta = e^\delta$ where expectancy $\mathrm{E}(\delta) \sim 0$ and variance $D(\delta) \sim \varepsilon^2$. For the sequential small mixtures (with independent $\beta_i$), we obtain the ratio

$$\frac{\alpha}{\eta_1 \cdot \eta_2 \cdot \ldots \cdot \eta_m} = \frac{\alpha}{e^{-\sum_i^m \delta_i}} \tag{1}$$

However, under the mild conditions, the distribution of $\sum_i^m \delta_i$ rapidly converges to a Gaussian distribution $\mathcal{N}(a, s^2)$ with $a \sim 0$ and $s \sim \sqrt{\sum_i^m \varepsilon_i^2}$. Thus, the limit pdf for sequential *arbitrary, but small* mixtures can be approximated by

$$f_\theta(x) = \int\limits_0^\infty \frac{1}{t\sigma\sqrt{2\pi}} \exp\left(-\frac{1}{2}\frac{x^2}{t^2\sigma^2}\right) \frac{1}{ts\sqrt{2\pi}} \exp\left[-\frac{(\ln t - a)^2}{2s^2}\right] dt, \qquad a \sim 0 \tag{2}$$

If we rescale $x \mapsto \frac{x}{\sigma}$ in this expression, then we obtain the representation for $f_\theta\left(\frac{x}{\sigma}\right)$; we also apply the change of variables $t \mapsto \sigma t = u$ and derive another formula of $f_\theta(x)$ in terms of $\mathcal{N}(0,1)$:

$$f_\theta(x) = \int_0^\infty \frac{1}{t\sigma\sqrt{2\pi}} \exp\left(-\frac{1}{2}\frac{x^2}{t^2\sigma^2}\right) \frac{1}{ts\sqrt{2\pi}} \exp\left(-\frac{\ln^2 t}{2s^2}\right) dt \qquad (3)$$

for some suitable parameters $s$ and $\sigma$.

Of course, the model of successive repeated small random perturbations applied to a stationary process is only one of many possible ones. However, this model allows statistical estimation of residuals in a simple and straightforward computational procedure and eventually allows the model data to be simulated in order to compare them with the actually observed data.

### 3. Method: Real Data Analysis

As in [1], we consider the absolute scalar data acquired by two of the Swarm satellites (Satellites Alpha and Bravo) at quasi-latitudes ranging between $+55°$ and $-55°$, and computed residuals with respect to the so-called VFM model of [7]: for the the array $ST_1$ of one-day std of residuals, take a look at Figure 1 borrowed from our article. Rescale this array $ST_1$ as $r \mapsto \frac{\sigma_1}{r} = y$ where $\sigma_1 = \text{mean}(ST_1)$ and $r \in ST_1$; by virtue of Equation (2), the array $\{y_i\}$ is expected to obey the lognormal distribution with parameter $s_1$ (see Equation (3)) and we can directly calculate $\sigma_1$ and $s_1$.

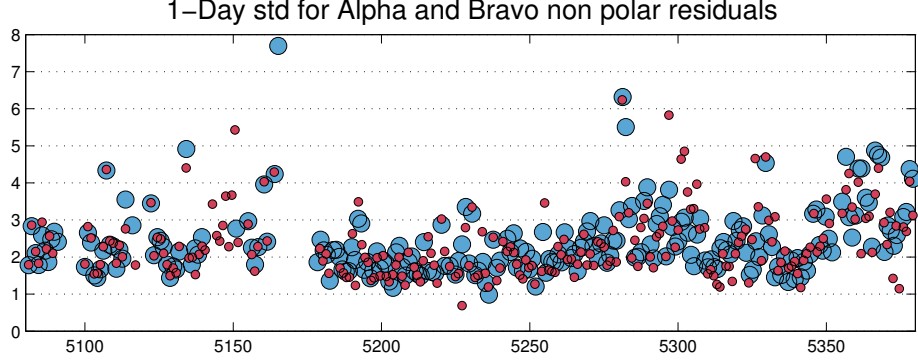

**Figure 1.** Standard deviations (in nT) computed every day for the mid-latitude residuals of the Swarm scalar data used to compute the VFM model of Vigneron et al. (2015). Blue large dots: data from the Swarm Alpha satellite and red dots: data from the Swarm Bravo satellite. Days are counted in Julian days, with 1 January 2000 taken as the reference.

Now, repeat all these computations for arrays $ST_{0.25}$, $ST_{0.5}$, $ST_{0.75}$ (i.e., corresponding to time intervals of 0.25 to 0.75 days). The results are as follows:

$ST_1$: Satellite A $\sigma_1 = 2.41$ $s_1 = 0.33$, Satellite B $\sigma_1 = 2.40$ $s_1 = 0.36$

$ST_{0.75}$: Satellite A $\sigma_{0.75} = 2.34$ $s_{0.75} = 0.36$, Satellite B $\sigma_{0.75} = 2.35$ $s_{0.75} = 0.39$

$ST_{0.5}$: Satellite A $\sigma_{0.5} = 2.22$ $s_{0.5} = 0.39$, Satellite B $\sigma_{0.5} = 2.21$ $s_{0.5} = 0.46$

$ST_{0.25}$: Satellite A $\sigma_{0.25} = 1.99$ $s_{0.25} = 0.43$, Satellite B $\sigma_{0.25} = 1.96$ $s_{0.25} = 0.49$

As often happens, a limited amount of lognormal data cannot provide stable statistical estimates. So, what are the "true values" of $s$ and $\sigma$? To answer this question, let us use the following well-known method of the statistical moments of $\theta$, namely:

$$
\begin{aligned}
\mathrm{E}|\theta| &= \int_{-\infty}^{+\infty} |x| f_\theta(x) dx = \int_0^\infty \left[ \int_{-\infty}^{+\infty} |x| \frac{1}{t\sigma\sqrt{2\pi}} \exp\left( -\frac{1}{2} \frac{x^2}{t^2\sigma^2} \right) dx \right] \cdot \frac{1}{ts\sqrt{2\pi}} \exp\left( -\frac{\ln^2 t}{2s^2} \right) ] dt \\
&= \int_0^\infty \sigma\sqrt{\frac{2}{\pi}} \cdot t \frac{1}{ts\sqrt{2\pi}} \exp\left( -\frac{\ln^2 t}{2s^2} \right) dt = \sigma\sqrt{\frac{2}{\pi}} \mathrm{E}\beta = \sigma\sqrt{\frac{2}{\pi}} e^{s^2/2} \\
\mathrm{E}\theta^2 &= \int_{-\infty}^{+\infty} x^2 f_\theta(x) dx = \int_0^\infty \mathrm{E}\alpha^2 \cdot t^2 \frac{1}{ts\sqrt{2\pi}} \exp\left( -\frac{\ln^2 t}{2s^2} \right) dt = \sigma^2 \mathrm{E}\beta^2 = \sigma^2 e^{2s^2}
\end{aligned}
$$

Thus, we obtain the explicit expressions of the unknown parameters

$$
\begin{cases}
s^2 = & \ln\frac{2}{\pi} + \ln \mathrm{E}\theta^2 - 2\ln \mathrm{E}|\theta| \\
\sigma^2 = & \frac{\pi^2}{4} \cdot \frac{(\mathrm{E}|\theta|)^4}{\mathrm{E}\theta^2}
\end{cases}
\tag{4}
$$

In practice, we recover from real data the estimates of the moments $\mathrm{E}\theta^2$, $\mathrm{E}|\theta|$ and then get (the estimates of) the unknown parameters.

The satellite scalar data [7] cover a little less than a year (between 29 November 2013 and 25 September 2014) and were further selected following a number of criteria, among which magnetically quiet and night-time conditions, to ensure that as little as possible non-modeled external signal is included in the data. This resulted in 42,160 data points for the Alpha satellite and 42,175 for the Bravo satellite. These data can be expected to reflect the signal of the field of internal origin the model tries to model, any other source of signal being treated as a source of noise acting on top of the very low instrumental and satellite noise (less than 0.3 nT, see [8–10]).

We may now add the quantitative details of the data distribution to the qualitative analysis of it that was published in [1]: namely, using formula (4), we may now recover the estimates of the parameters $s$ and $\sigma$ (the latter can be treated as an estimate for the "inner precision" of measurements); Figure 2 actually confirms the fact that this close-to-Laplacian distribution indeed can be represented as the result of a lognormal mixture according to formula (3).

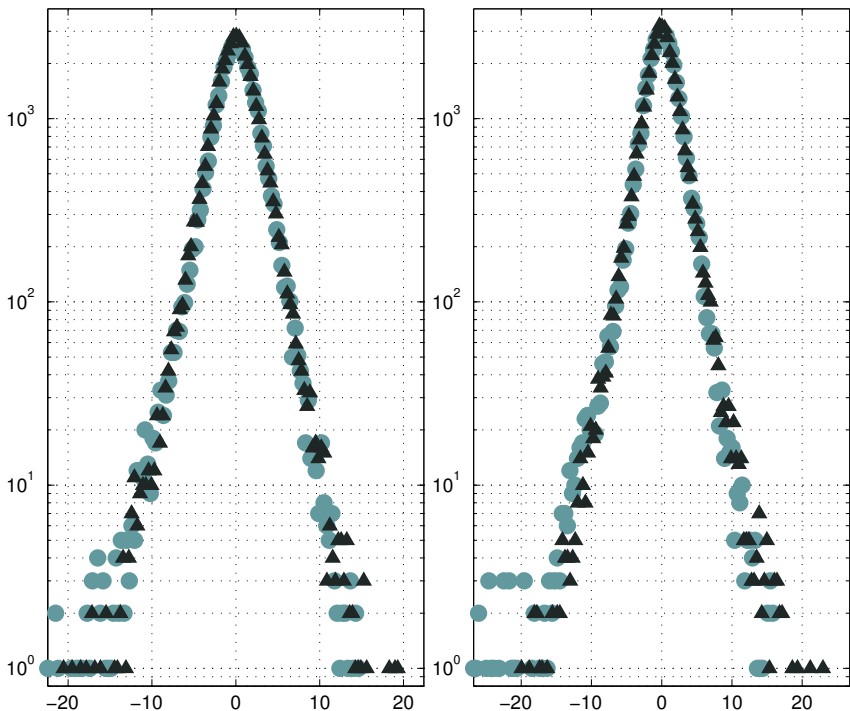

**Figure 2.** Left: Histogram of the residuals (circles) of the Swarm Alpha scalar data used to compute the VFM model [7] together with histogram (triangles) of an identical number of simulated mixture of Gaussian distributions according to the parameters $s = 0.41$, $\sigma = 2.18$ recovered from the real data; right: the same plots but for the Swarm Bravo scalar data, parameters $s = 0.47$ and $\sigma = 2.10$.

## 4. Discussion

In general, the study of the behavior of residuals in the case of magnetic observations is aimed at quantifying the accuracy of instrumental measurements and to quantify external influences, which are calculated differently in the Gaussian data model and in the mixed model.

As part of the reasoning for the applicability of the aforementioned mixture model in case of SWARM data, it appears that for moderate amounts of data ($N \sim 10^3$), the KS test does not reject the model; nevertheless, as $N$ grows to several tens of thousands measurements, the model clearly does not exhaust the complexity of inhomogeneous effects available in the SWARM magnetic data and is therefore formally rejected by the KS criterion.

An important detail related to our model is that we assume a continuous mixture and obtain its analytic form. This is certainly an idealistic assumption, which needs not be absolutely true, especially if we have strong discrete effects in the magnetic data: for example, if we consider daytime and nighttime observations simultaneously. In this case, we are better off using a probabilistic model, which assumes that all data points appear from a mixture of a finite number of Gaussian distributions with unknown parameters. Fortunately, when the number of Gaussian distributions involved is known (at least approximately), then a well-known EM algorithm can provide (via maximum likelihood estimation) the parameters of the corresponding discrete statistical model.

## 5. Conclusions

This purely numerical approach is not new, but in the present study, we consider an assumption that seems to be a rather novel approach to practical residuals. Which approach is more appropriate depends on the specific situation. For example, the continuous analytical formula given above gives hope that in the case of ground-based magnetic observations over a relatively short time interval, our methods for calculating the unsteadiness will

prove to be a useful refinement and, at the same time, allow us to isolate subtle effects of unsteady field behavior.

**Funding:** This work was conducted in the framework of budgetary funding of the Geophysical Center of Russian Academy of Sciences, adopted by the Ministry of Science and Higher Education of the Russian Federation.

**Data Availability Statement:** The data used in this study have already been published in [1,7].

**Conflicts of Interest:** The author declares no conflict of interest.

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
