# Peer review of "Processing Non-Gaussian Data Residuals in Geomagnetism"

_applsci, doi:10.3390/app12042097_

Round 1
Reviewer 1 Report
The identification of different sources of noise/errors in geomagnetic data is of critical value to assess the resulting products from the data viz. the global models. Thus research presented in this manuscript is important.
In the Section on Real Data Analysis, the treatment of the error distribution of data from Swarm A and B has been done on night time data for quiet days. This analysis needs to be expanded to show the sources of the errors of Gaussian and mixed models from data of disturbed days also to establish the efficacy of the approach.
I also suggest that the changes of the model with increasing amounts/lengths of data should be demonstrated. This would increase the impact of the manuscript.
Author Response
In the Section on Real Data Analysis, the treatment of the error distribution of data from Swarm A and B
has been done on night time data for quiet days. This analysis needs to be expanded to show the
sources of the errors of Gaussian and mixed models from data of disturbed days also to establish the
efficacy of the approach.
Yes, I completely agree with this suggestion. My answer: the data model may not be a continuous mixture of
Gaussians, but e.g., be a discrete mixture of them; this means mixing a finite number of Gaussian observables
with different parameters.
In this case, the mixture can be numerically decomposed using the well-known EM algorithm (part of the
collection of of machine learning methods). In my manuscript, I considered a different case of the data and its
corresponding analytical solution.
I have added a further explanation in Section 4, “Discussion and Conclusion” (lines 132-143).
I also suggest that the changes of the model with increasing amounts/lengths of data should be
demonstrated. This would increase the impact of the manuscript.
I only partially agree with this suggestion because the data in question are not from a ground-based magnetic
observatory, so their overly detailed statistical descriptions seem beyond the scope of the main topic.
Nevertheless, I added the following sentence (lines 132-134) to the text of the article
"An important detail related to our model is that we assume a continuous mixture
and obtain its analytic form. This is certainly an idealistic assumption, which need not be absolutely true..."

Reviewer 2 Report
Report:
The work entitled "Processing the non-Gaussian data residuals in geomagnetism" by Andrey Khokhlov.
The article is well-structured that could be published after some observations and clarify some questions.
The author clearly shows the objective in terms of rigorous data processing, especially when the random variables have a local average behavior that differs significantly from the global average. Thus, it shows the need to use a Gaussian sum instead of a single one, considering the possible different origins of the disturbances of the different variables.
Author Response
The article is well-structured that could be published after some observations and clarify some questions.
Indeed, I decided to clarify the case when the data model may not be a continuous mixture of Gaussians, but e.g., be a discrete mixture of them; this means mixing a finite number of Gaussian observables with different parameters. The following text was added to Section 4, “Discussion and Conclusion” (lines 132-147).
An important detail related to our model is that we assume a continuous mixtureand obtain its analytic form. This is certainly an idealistic assumption, which need notbe absolutely true, especially if we have strong discrete effects in the magnetic data: for example, if we consider daytime and nighttime observations simultaneously. In this case we are better off using a probabilistic model, which assumes that all data points appear from a mixture of a finite number of Gaussian distributions with unknown parameters. Fortunately, when the number of Gaussian distributions involved is known(at least approximately), then a well-known EM algorithm can provide (via maximum likelihood estimation) the parameters of the corresponding discrete statistical model.This purely numerical approach is not new, but in the present study we consider an assumption that seems to be a rather novel approach to practical residuals.
Which approach is more appropriate depends on the specific situation. For example, the continuous analytical formula given above gives hope that in the case of ground-based magnetic observations over a relatively short time interval, our methods for calculating the unsteadiness will prove to be a useful refinement and at the same time allow us to isolate subtle effects of unsteady field behavior.